Italian and Swedish adolescents: differences and associations in subjective well-being and psychological well-being

Garcia Danilo danilo.garcia@icloud.com 1 2 3 4
Sagone Elisabetta 5
De Caroli Maria Elvira 5
Nima Ali Al 1 2 3
1 Blekinge Center of Competence, Blekinge County Council , Karlskrona , Sweden
2 Department of Psychology, University of Gothenburg , Gothenburg , Sweden
3 Network for Empowerment and Well-Being , Sweden
4 Department of Psychology, Lund University , Lund , Sweden
5 Department of Educational Sciences, University of Catania , Catania , Sicily , Italy
D’Acquisto Fulvio
Electronic publication date: 2017 Jan 12
Publication date: 2017
Volume: 5
Electronic Location ID: e2868
Received 2016 Aug 25; Accepted 2016 Dec 6
Copyright: ©2017 Garcia et al.
Copyright year: 2017
Copyright holder: Garcia et al.
License: This is an open access article distributed under the terms of the Creative Commons Attribution License, which permits unrestricted use, distribution, reproduction and adaptation in any medium and for any purpose provided that it is properly attributed. For attribution, the original author(s), title, publication source (PeerJ) and either DOI or URL of the article must be cited.
License URL: https://creativecommons.org/licenses/by/4.0/

Keywords: Italy, Adolescents, Sweden, Life satisfaction, Negative Affect, Psychological well-being, Positive Affect, Subjective Well-Being

Funding: FINSAM Sustainable Personal Development The data collection among Swedish Adolescents was funded by FINSAM. This data is part of the research project Sustainable Personal Development at Blekinge Centre of Competence. The funders had no role in study design, data collection and analysis, decision to publish, or preparation of the manuscript.

==============================
Background

One important aspect of subjective judgments about one’s well-being (i.e., subjective well-being: life satisfaction, positive affect, and negative affect) is that cultural features, such as, nationality seem to shape cognitive judgments about the “the ideal life.” In this comparative study we examined differences in subjective well-being and psychological well-being between Italian and Swedish adolescents and tested if the relationship between the three constructs of subjective well-being (i.e., satisfaction with life, positive affect, and negative affect) and psychological well-being was moderated by the adolescents’ nationality.

Method

Italian (n = 255) and Swedish (n = 277) adolescents answered to the Satisfaction with Life Scale, the Positive Affect Negative Affect Schedule, and Ryff’s Scales of Psychological Well-Being. Differences between samples were tested using a Multiple Analysis of Variance. We also conducted a multiple group analysis (Italy and Sweden) using Structural Equation Modelling to investigate the relationship between all three subjective well-being constructs and psychological well-being.

Results

Italian adolescents scored significantly higher in satisfaction with life than Swedish adolescents. Additionally, across countries, girls scored significantly higher in negative affect than boys. In both countries, all three constructs of subjective well-being were significantly associated to adolescents’ psychological well-being. Nevertheless, while the effect of the relationship between affect and psychological well-being was almost the same across countries, life satisfaction was more strongly related to psychological well-being among Swedish adolescents.

Conclusions

The present study shows that there are larger variations between these two cultures in the cognitive construct of subjective well-being than in the affective construct. Accordingly, associations between the cognitive component, not the affective component, of subjective well-being and psychological well-being differ between countries as well.

Introduction

Researchers in the field of Positive Psychology (Seligman, 2002) and the Science of Well-Being (Cloninger, 2004; Cloninger & Garcia, 2015) are interested in protective factors of healthy human development. Among these factors, both subjective well-being and psychological well-being1 are considered as interrelated psychological characteristics involved in positive human functioning, such as, resilience and hardiness (Seligman & Csikszentmihalyi, 2014; Fredrickson, 2001; Folkman & Moskowitz, 2000). Subjective well-being is composed of life satisfaction, positive affect, and negative affect (Diener, 1984). Life satisfaction entails the cognitive part of subjective well-being and is the person’s evaluation of her/his life as whole in relation to her/his ideal life (Diener, 1984). Judgments of life satisfaction are influenced by information accessible at the time of the evaluation (see Schimmack, Diener & Oishi, 2002). Positive affect entails a person’s tendency to feel positive states, such as, enthusiastic, active, and alert, while negative affect the tendency to feel distress and unpleasurable engagement, such as, anger, disgust, guilt, and fear (Watson, Clark & Tellegen, 1988). Swedish adolescents characterized by high levels of positive affect in combination with low levels of negative affect (i.e., a self-fulfilling affective profile) experience high levels of energy, optimism, self-esteem, optimism, internal locus of control, and low levels of stress (Archer et al., 2007; Archer, Adolfsson & Karlsson, 2008). More recently, among Italian adolescents, Di Fabio & Bucci (2015) demonstrated that high school students with a self-fulfilling profile scored higher on life satisfaction, self-esteem, life meaning, and optimism than students with any other type of affective profile (see Garcia, 2011 for similar results among Swedish adolescents). Thus, showing that the affective construct of subjective well-being is associated to similar positive outcomes across Swedish and Italian cultures during adolescence.

Psychological well-being is also considered a major factor for optimal human functioning (Ryff & Keyes, 1995; Ryff & Singer, 1998), occasionally investigated as a predictor variable or an antecedent of individual positive development, but also as the outcome of high levels of subjective well-being (Ryff, 2013). The elements of psychological well-being are represented by self-acceptance (i.e., acceptance of the self, self-actualization, optimal functioning, and maturity), positive relations with the others (i.e., the ability to express feelings of empathy and affection for all human beings and to be able of greater love and friendship, and identification with others), autonomy (i.e., independence and regulation of behavior through internal locus of control), environmental mastery (i.e., the ability to create environments suitable to healthy conditions), purpose in life (i.e., a sense of goal directedness and intentionality), and, finally, personal growth (i.e., the realization of one’s potentialities, underlining the value of new challenges at different moments of one’s own life). Psychological well-being has been studied among adolescents in relation to other constructs such as resilience and hardiness (Kobasa, Maddi & Kahn, 1982; Masten et al., 1999; Sagone & De Caroli, 2014; De Caroli & Sagone, 2016), adaptive coping strategies and sense of coherence (Pallant & Lae, 2002), and in relation to subjective well-being operationalized as affective profiles (Garcia & Siddiqui, 2009a; Garcia & Siddiqui, 2009b; Garcia, Nima & Kjell, 2014).

Among Swedish adolescents and young adults, for example, psychological well-being has been associated to all three constructs of subjective well-being, essentially showing that high levels of positive affect, low levels of negative affect, and high levels of life satisfaction are strongly related to high levels of psychological well-being (Garcia & Siddiqui, 2009a; Garcia & Siddiqui, 2009b; Garcia & Archer, 2012; Garcia, Nima & Kjell, 2014). Accordingly, more recently, among Italian adolescents, it was noted that individuals with the self-fulfilling profile reported higher resilience (in detail, sense of humor, competence, adaptability, and engagement) and psychological well-being (in terms of full autonomy, sense of purpose in life, and self-acceptance) than adolescents with any other type of affective profile (De Caroli & Sagone, 2016). Furthermore, Italian boys expressed higher levels of psychological well-being than girls, specifically, in environmental mastery and self-acceptance (Sagone & De Caroli, 2014. See Garcia, 2011 for similar results among Swedish adolescents). In some other studies, psychological well-being has showed significant correlations to other measures of well-being, such as, sense of vitality (Ryan & Frederick, 1997) and optimism (Scheier, Carver & Bridges, 2001).

One important aspect of the subjective well-being constructs (i.e., life satisfaction, positive affect, and negative affect) is that cultural features, such as, nationality, ethnicity, religious affiliation, and motivation seem to shape how individuals understand “the ideal life” (Tsai, Knutson & Fung, 2006; Tsai, Miao & Seppala, 2007; Scollon et al., 2009). In other words, we could expect larger variations between cultures in the cognitive construct of subjective well-being than in the affective construct. If so, associations between the subjective well-being constructs and psychological well-being might differ between countries as well.

The present study

The study of well-being during adolescence is important since this period of life is characterized by various events and transitions that significantly influence adolescents’ well-being (Kjell et al., 2013). Moreover, although research on adolescents’ well-being has increased in the last decade (e.g., see Garcia & Siddiqui, 2009a; Garcia & Siddiqui, 2009b; Garcia & Sikström, 2013; Fogle, Huebner & Laughlin, 2002; Funk III, Huebner & Valois, 2006; Sagone & De Caroli, 2014; De Caroli & Sagone, 2016), the study of well-being across cultures is still scarce (for a recent review showing that the majority of previous research in this area involves American participants see Proctor, Linley & Maltby, 2009). The main purposes of this study were (1) to analyze differences in subjective well-being and psychological well-being between Italian and Swedish adolescents and (2) to test if the relationships between the three aspects of subjective well-being (i.e., satisfaction with life, positive affect, and negative affect) and psychological well-being were moderated by the adolescents’ nationality.

Method

Ethical statement

After consulting with the university’s Network for Empowerment and Well-Being’s Review Board and according to law (2003: 460, section 2) concerning the ethical research involving humans we arrived at the conclusion that the design of the present study (e.g., all participants’ data were anonymous and will not be used for commercial or other non-scientific purposes) required only verbal consent from participants. For the Italian sample, researchers followed the Ethical Code for Italian psychologists (L. 18.02.1989, n.56) and DL for data privacy (DLGS 196/2003); Ethical Code for Psychological Research (March 27, 2015) by AIP (Italian Psychologists Association). For the Italian sample also only verbal consent was needed.

Participants

The data was collected at two high schools in Eastern Sicily, Italy (N = 255, 107 boys and 148 girls, mean age = 16.19 years SD = 1.75 years) and two high schools in the West of Sweden (N = 277, 166 boys and 111 girls, mean age 18.11 years SD 0.59 years). The sampling procedure of schools was based on convenience. Teachers and parents were informed about the nature of the study. The school nurse from each school was contacted by the researchers and informed about the study in case any of the students needed counseling. Participants were informed that the study examined how pupils think about their lives in different situations. They were ensured anonymity and informed that participation was voluntary; they had consent from their teachers to participate. The study was conducted in the participants’ own classrooms in groups of 20–30 pupils; the questionnaires were distributed on paper. The entire procedure, including debriefing, took approximately 30 min.

Measures

The Satisfaction With Life Scale (Pavot & Diener, 2008) assesses the cognitive component of subjective well-being (i.e., life satisfaction) and consists of five items (e.g., “In most of my ways my life is close to my ideal”) that require a response on a 7-point Likert scale (1 = strongly disagree, 7 = strongly agree). Both the Swedish and the Italian versions of this instrument have been previously used in these cultures (e.g., Garcia & Siddiqui, 2009a; Garcia & Siddiqui, 2009b; Fahlgren et al., 2015; Sagone & De Caroli, 2015). In the current study, this measure had a Cronbach’s α = .85 in the Italian sample and .93 in the Swedish sample.

The Positive Affect and Negative Affect Schedule (Watson, Clark & Tellegen, 1988) assesses the affective component of subjective well-being by requiring participants to indicate on 5-point Likert scale to what extent (1 = very slightly, 5 = extremely) they generally experienced 20 adjectives describing affective states (10 for positive affect and 10 for negative affect) within the last few weeks. The positive affect scale includes adjectives such as “strong,” “proud,” and “interested”; and the negative affect scale includes adjectives such as “afraid,” “ashamed,” and “nervous.” The Swedish and Italian versions have been used in previous studies (e.g., Schütz, Archer & Garcia, 2013; De Caroli & Sagone, 2016) and demonstrated acceptable internal consistency in the present study: Cronbach’s α was .77 for positive affect and .79 for negative affect in the Italian sample and .86 for positive affect and .85 for negative affect in the Swedish sample.

Ryff’s Scales of Psychological Well-Being—short version (Clarke et al., 2001) comprises 18 items with a 6-point Likert (1 = strongly disagree, 6 = strongly agree), three items for each of the six dimensions: self-acceptance (e.g., “I like most aspects of my personality”), personal growth (e.g., “For me, life has been a continuous process of learning, changing, and growth”), purpose in life (“Some people wander aimlessly through life, but I am not one of them”), environmental mastery (e.g., “I am quite good at managing the responsibilities of my daily life”), autonomy (e.g., “I have confidence in my own opinions, even if they are contrary to the general consensus”), and positive relations with others (e.g., “People would describe me as a giving person, willing to share my time with others”). The Swedish and Italian versions have been used in previous studies (e.g., Garcia et al., 2015; De Caroli & Sagone, 2016). In the present study, we used the whole scale as one general measure of psychological well-being. The Cronbach’s α for this psychological well-being composite score were .68 for the Italian sample and .79 for the Swedish sample.

Results

We conducted one Multivariate Analysis of Variance using age as covariate in order to investigate differences between Italian and Swedish adolescents. Specifically, we used country (Italy–Sweden) and gender (male–female) as the independent factors, age as the covariate, and the different constructs of subjective well-being (satisfaction with life, positive affect, and negative affect) and the psychological well-being composite as the independent variables. We used age as a covariate since the significant difference in age between samples (age meanItaly = 16.19 ± 1.75, age meanSweden = 18.11 ± 0.59) and the fact that levels of affectivity and psychological well-being fluctuate with age (see Ryff, 1989).

Both gender (F(4, 519) = 4.84; p < .01, Wilks’ Lambda = .96) and country (F(4, 519) = 3.49; p < .01, Wilks’ Lambda = .97) had a significant effect on satisfaction with life and negative affect. Italian adolescents (M = 4.62, SD = 1.26) scored significantly higher in satisfaction with life (F(1, 522) = 6.85; p < .01) than Swedish adolescents (M = 4.30, SD = 1.56). Additionally, girls (M = 2.38, SD = .67) scored significantly higher in negative affect (F(1, 522) = 13.75; p < .001) than boys (M = 2.17, SD = .62). The interaction of country and gender was no significant (F(4, 519) = .76; p = .552, Wilks’ Lambda = .99), thus, the difference in life satisfaction between countries was consistent across genders and the difference in negative affect between girls and boys was consistent across countries. No other significant differences were found.

The second analysis was a multiple group Structural Equation modeling to test if the relationship between the three constructs of subjective well-being (i.e., satisfaction with life, positive affect, and negative affect) and psychological well-being was moderated by individuals’ nationality. In other words, we used country as the moderator, all three subjective well-being constructs as the independent variables, and psychological well-being as the dependent variable. This model showed a goodness of fit index = 1.00, a comparative fit index = 1.00, an incremental fit index = 1.00, and normed fit index = 1.00. Thus, indicating that the model is a good-fitting model.

As showed in Fig. 1 and Table 1, in both countries all three constructs of subjective well-being were associated to adolescents’ psychological well-being. We tested if the regression weights were significantly different across groups (i.e., both Italian and Swedish adolescents). The result showed that there was a significant difference (Chi2 = 9.53, p < .05) on level model between the unconstrained/original model and constrained model (with all constrained regression weights on psychological well-being). In other words, nationality moderated at least one of the paths. In further analyses (path by path), we found that the effect of the relationships between positive affect and psychological well-being (Chi2 = 1.24, p = .27) and between negative affect and psychological well-being (Chi2 = 0.17, p = .68) were not significantly different between countries. The relationship between life satisfaction and psychological well-being, on the other hand, differed significantly between countries (Chi2 = 4.12, p < .05). This relationship was significantly stronger among Swedish adolescents (see Table 2).

Figure 1 Structural equation modelling showing all correlations for (A) Italian (N = 255) and (B) Swedish adolescents’ (N = 275) subjective well-being and psychological well-being, along the standardized parameter estimates.

Table 1 Results of the structural equation model using all three subjective well-being constructs as the predictors of (A) Italian (N = 255) and (B) Swedish adolescents’ (N = 275) psychological well-being.

Predictor	Outcome	β	SE	B	P	
A. Italian adolescents	
Satisfaction with life	Psychological well-being R2 = .41	.15	.02	.06	<.001	
Positive affect	.35	.05	.30	<.001	
Negative affect	−.34	.04	−.28	<.001	
B. Swedish adolescents	
Satisfaction with life		.30	.02	.12	<.001	
Positive affect	Psychological well-being R2 = .57	.39	.04	.37	<.001	
Negative affect		−.29	.04	−.26	<.001	

Table 2 Results of the compare nested models using Chi-square difference test.

Model	DF	Chi2 difference	P	
All constrained regression weights on psychological well-being	3	9.53	.02	
Constrained regression weight satisfaction with life on psychological well-being	1	4.12	.04	
Constrained regression weights positive affect on psychological well-being	1	1.24	.27	
Constrained regression weights negative affect on psychological well-being	1	.17	.68	
Notes.

Unconstrained model has Chi2 = 0.00, df = 0.00.

Discussion

The purpose of this study was twofold: (1) to analyze differences in subjective well-being and psychological well-being between Italian and Swedish adolescents and (2) to test if the relationships between the three aspects of subjective well-being (i.e., satisfaction with life, positive affect, and negative affect) and psychological well-being were moderated by the adolescents’ nationality. At a general level, the results were straightforward: Italian adolescents experience their life as highly satisfying and more close to their ideal than Swedish adolescents, and girls in both countries experience more negative emotions than boys. No significant differences were found with regard to positive emotions or psychological well-being.

The gender differences here have actually been found earlier and are substantially supported across the literature. Females usually score higher in characteristics related to negative emotionality, such as, neuroticism, anxiety, depression, rumination, and score lower in dispositional optimism than males (see among others Ellis, 2011; Hyde, Mezulis & Abramson, 2008; Johnson & Whisman, 2013; Twenge & Nolen-Hoeksema, 2002; Bodas & Ollendick, 2005; De Bolle et al., 2015; Hopcroft & McLaughlin, 2012; McCrae et al., 2005; Sagone & De Caroli, 2016). Interestingly, gender differences in neuroticism, which is almost synonymous with negative affect (Tellegen, 1993), do not reach their full strength until around age 14 (De Bolle et al., 2015); which was the age of the youngest participants in the present study. That being said, although gender differences in negative affect states and traits are present across nations in most studies, these differences seem to depend on which level of gender equity is practiced (Schmitt et al., 2016). Although counter-intuitive, gender differences in negative emotionality are larger in relatively high gender egalitarian cultures (Schmitt et al., 2016). Since we did not found any interactions between gender and nationality, our results implicitly suggest that Italy and Sweden might be relatively alike in gender equality. For instance, although Sweden has the fourth place and Italy the fiftieth place in the 2016 report from the World Economic Forum’s Global Gender Gap, the mean score of both countries in the area of educational attainment (including the gender equity measured by the Education Indicators by UNESCO) is almost the same (for Sweden 0.999 and for Italy 0.995; see https://www.weforum.org/reports/the-global-gender-gap-report-2016). This is, however, beyond the scope of the present study and we suggest it as an interesting venue for future studies. Additionally, although we state that cultural features, such as nationality, ethnicity, religious affiliation, and motivation seem to shape how individuals understand “the ideal life” (Tsai, Knutson & Fung, 2006; Tsai, Miao & Seppala, 2007; Scollon et al., 2009. See also George, Ellison & Larson, 2002; Green & Elliott, 2009), we do not address, for example, how religious affiliations or “ideal life” constructions relate to our findings. This is definitely another venue for further exploration.

We did also found that affectivity (both positive and negative affect) is equally related to psychological well-being across both nations, while life satisfaction was significantly more strongly related to psychological well-being among Swedish adolescents than among Italian adolescents. Firstly, this demonstrated how affectivity and life satisfaction are different constructs of subjective well-being (cf. Diener, 1984)—one addressing an affective part (i.e., emotions) and the other a cognitive part (i.e., life satisfaction). That is, a person cognitions or self-assessment of biological states, such emotions, and psychological phenomena, such as, life satisfaction. That being said, in order to have a biopsychosocial model of subjective well-being, we lacked a social component. It is plausible that such a construct is differently associated to psychological well-being among different cultures (cf. Markus & Kitayama, 1991). Future studies should investigate this further. Harmony, for example, has recently being suggested as a cognitive construct that complements life satisfaction (Kjell et al., 2016). However, Garcia (in press) goes even further and points out that since harmony is a construct related to the sense of balance and flexibility that an individual experiences in relation to the world or her/his life (cf. Li, 2008a; Li, 2008b), harmony is more likely a social component of subjective well-being. Subjective well-being, in turn, is more of a cognitive global construct of well-being, since it is mostly measured through self-reports (see Garcia, in press, who suggest affect as the cognitive biological part of subjective well-being, life satisfaction as the cognitive psychological part, and harmony as the cognitive social part). Secondly, the differences in associations between life satisfaction and psychological well-being among adolescents from these two countries are in line with the fact that cultural features seem to shape how individuals understand “the ideal life” (Tsai, Knutson & Fung, 2006; Tsai, Miao & Seppala, 2007; Scollon et al., 2009).

Limitations and final remarks

The cross-sectional and self-report design of the present study limit our conclusions. Also, it is plausible that the difference in age between the two groups could be the factor driving the between-groups difference, rather than nationality. For instance, this could also be the cause of different reliability coefficients for the different measures used here. Additionally, a more valid measure of psychological well-being would allowed us to scrutinize the sub-scales. The different aspects of psychological well-being like self-acceptance, purpose in life and/or positive relations with others might vary between adolescents from Sweden and Italy (cf. Markus & Kitayama, 1991). That being said, the present study shows that there are larger variations between these two cultures in the cognitive construct of subjective well-being than in the affective construct. Accordingly, associations between the cognitive component, not the affective component, of subjective well-being and psychological well-being differ between countries as well.

Supplemental Information

Data S1 Raw Data

Click here for additional data file.

Additional Information and Declarations

Competing Interests

Author Contributions

Human Ethics

Data Availability

1 For an interesting debate on whether wellness equals happiness, and suggesting that subjective well-being is the ‘Big One’ appropriate assessment of both see among others: Biswas-Diener, Kashdan & King, 2009; Delle Fave & Bassi, 2009; Kashdan, Biswas-Diener & King, 2008; Ryan & Huta, 2009; Straume & Vittersø, 2012; Waterman, 2008.

Dr. Danilo Garcia is the Director of the Blekinge Center of Competence, which is the Blekinge County Council’s research and development unit. The Center works on innovations in public health and practice through interdisciplinary scientific research, person-centered methods, community projects, and the dissemination of knowledge in order to increase the quality of life of the habitants of the county of Blekinge, Sweden. He is also an Associate Professor at the University of Gothenburg and together with Professor Trevor Archer and Associate Professor Max Rapp Ricciardi, the leading researcher of the Network for Empowerment and Well-Being. Ali Al Nima is a researcher and statistician at the Blekinge Center of Competence and a member of the Network for Empowerment and Well-Being.

Danilo Garcia conceived and designed the experiments, performed the experiments, analyzed the data, contributed reagents/materials/analysis tools, wrote the paper, prepared figures and/or tables, reviewed drafts of the paper.

Elisabetta Sagone conceived and designed the experiments, performed the experiments, wrote the paper, reviewed drafts of the paper.

Maria Elvira De Caroli performed the experiments, reviewed drafts of the paper.

Ali Al Nima performed the experiments, analyzed the data, contributed reagents/materials/analysis tools, prepared figures and/or tables, reviewed drafts of the paper.

The following information was supplied relating to ethical approvals (i.e., approving body and any reference numbers):

After consulting with the university’s Network for Empowerment and Well-Being’s Review Board and according to law (2003: 460, section 2) concerning the ethical research involving humans we arrived at the conclusion that the design of the present study (e.g., all participants’ data were anonymous and will not be used for commercial or other non-scientific purposes) required only informed verbal from participants. For the Italian sample, researchers followed the Ethical Code for Italian psychologists (L. 18.02.1989, n.56) and DL for data privacy (DLGS 196/2003); Ethical Code for Psychological Research (March 27, 2015) by AIP (Italian Psychologists Association). For the Italian sample also only verbal consent was needed.

The following information was supplied regarding data availability:

The raw data has been supplied as a Data S1.

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
