# Peer review of "Italian and Swedish adolescents: differences and associations in subjective well-being and psychological well-being"

_PeerJ, doi:10.7717/peerj.2868_

## Round 0.1 · original submission · Minor Revisions

Please consider the comprehensive comments of the two reviewers. I suggest that you aim to include in the revised text as many reviewer suggestions as possible as this will certainly will improve the impact of the study.

·

Basic reporting

Please also see the attached .pdf file for very minor changes that might improve reading clarity.

Other minor points:

In the Results sub-section of the Abstract, it states that “…all three constructs were associated to adolescents’ psychological wellbeing…”. For clarity it may be worth stating that these were significant associations in all cases.

Line 207: “The relationship between life satisfaction and psychological wellbeing…differed significantly between countries” – It might help to include the relevant chi-square and p values from Table 2 at the end of this sentence, to illustrate the point.

References in the section from lines 122-127 are not clear, and do not contain full details (e.g. author or authoring body).

Figures 2 and 3 are clear and illustrative, so you could almost do without Figure 1 (as the information is contained within the subsequent figures).

It might be useful to either label ‘e1’ in the Figures, or explain what this factor is in the figure captions. (Is it nationality?)

Experimental design

Adolescent rationale

Given the use of adolescent samples, it would be worthwhile to highlight the importance of studying adolescents in the Introduction more prominently. Are previous associations between different aspects/components of wellbeing expected to differ between adolescents and adults? If so, how? If not, is there any particular impetus for considering the wellbeing of adolescents rather than another age group? Such effects are briefly mentioned in the first paragraph of the Results (lines 179-181), but this needs to be addressed in the Introduction.


Cross-cultural rationale

Lines 108-111 (also 268-270 in the Discussion): The rationale for looking for cross-cultural differences in cognitive components of wellbeing measures is sound, but it is less clear than the rationale for expecting such a cultural difference between Italian and Swedish samples (compared to the differences between Christian and Buddhist cultures highlighted by Tsai et al.). Is there any national census/cohort data (e.g. showing differences in relevant values between Italy and Sweden) or aspects of natural experiments (e.g. recent social/political unrest) that could be referred to that would justify expecting differences in wellbeing or values between these two European countries?


Gender effects

Given that there are sizeable and significant effects of gender reported in the Results section, it would be worth briefly discussing some background literature on gender differences in wellbeing in the Introduction.

In the Discussion (line 234) reference is made to earlier work to have found gender differences similar to those found in the current study. If this is the case, then such findings should be mentioned at some point in the Introduction – as this was pre-existing knowledge and not changed as a result of the findings of the current study.


Theoretical distinctions between wellbeing measures

The abstract refers to “3 subjective wellbeing constructs” and 1 psychological wellbeing measure. However, within the 3 subjective wellbeing constructs, 2 are from the same scale (positive affect and negative affect from the PANAS) and 1 is from the Satisfaction With Life Scale (SWLS). It might be clearer to therefore refer to 2 affect scales, satisfaction with life, and psychological wellbeing separately, rather than collecting the former 3 scales together. Otherwise, it would be advisable to briefly mention the justification for why the SWLS is grouped together with the 2 PANAS scales (i.e. the fact that this distinction is made in background literature) somewhere in the Introduction. Alternatively, it might be worth having in the Methods section some empirical support from the current dataset in regards to separating the questionnaire measures into subjective and psychological wellbeing - e.g. maybe alpha scores, or a very short reference to a factor analysis, or something similar.

Lines 62-65 would benefit from clarifying both why life satisfaction as a “cognitive” component of wellbeing is theoretically closer to affect than to psychological wellbeing. That is to say that the argument for separating the 4 measures (life satisfaction, the positive and negative sub-scales of the PANAS, and Ryff’s wellbeing scales) into 2 components (subjective and psychological wellbeing) could be clearer here – otherwise it might make more sense to just refer to the satisfaction with life scales, Ryff’s scale, and the 2 PANAS sub-scales separately throughout. In particular, since the results find a distinction between life satisfaction and positive/negative affect in the SEMs, these potential distinctions deserve greater attention in the Introduction.

Lines 256-257: The distinction made between emotions as being “more biological” than the “cognitive” other measures of the study seems quite strong, seeing as self-report measures were used in all cases in the current study. This point would be stronger if the authors had used physiological measures of e.g. arousal / skin conductance as a further index of emotionality in the study. Would the authors expect the same results to be found if the current study were replicated using physiological measures rather than self-report measures of affect? If not, it might be better to stick to the distinction that the authors use elsewhere in the study (i.e. between affective and cognitive constructs) rather than between explicitly biological and psychological.

Validity of the findings

Age differences

I have some concern about the difference in age between the two groups (could this be driving the between-groups difference, rather than nationality?)

Lines 129-131: It is a little concerning that the ages of the samples are quite different – roughly 2 years apart, especially in light of the very low standard deviation of the ages in the Swedish sample. It also seems a bit strange that if all participants were recruited from high schools, that the mean age of the Swedish sample is over 18 years. The inclusion of age as a (non-significant) covariate in the MANOVA is very encouraging, however the age difference between the samples needs to be addressed as a limitation in the Discussion section – not least because it is not clear why this age range (late adolescence) in particular is being looked at in the current study.
Furthermore, 18 (and to a lesser extent, 16) seems quite old to be considered “adolescent”. Adolescent is a term which (while variable in the age it refers to) is perhaps more commonly associated with 13-15 years. As such a term such as “older adolescents and young adults” might be more appropriate for the current study. Otherwise, there should be some justification in the Introduction for why this age group (16-18) is of particular interest to the current researchers.

Lines 155-158 and 170-172: It would be good to address (in the Discussion) the fact that the alpha scores for the PANAS and psychological wellbeing scale were different across the two samples – my first thought would be that this might be related to the different between the two groups in terms of variation in age.


State and trait measures

Generally throughout (especially when discussing past research) it would improve the clarity of the paper to specify whether previous effects (and those in the current study) are specific to either trait or state measures, or whether effects are comparable when using these two different classes of measures.

Lines 65-68 Refer to positive and negative affect as individuals’ tendencies to feel either positive or negative feelings, but the PANAS used in the current study is more of a state measure than a trait as implied here. There should be some mention of this limitation in the Discussion, or else these lines should be rephrased to reflect that the authors are more interested in state measures than “tendencies to feel states”.
Furthermore, I think it would be interesting for the authors to speculate on whether similar or diverse results would be expected if using more trait-like measures of either affect (e.g. neuroticism or something similar), and/or wellbeing.

Lines 235-251 refer to both state and trait measures, but the current study did not include any trait measures. This is a point that needs to be addressed somewhere in the Discussion as a limitation of the current study.


Age-appropriateness of measures

It might be worth providing some references to previous studies to have shown the measures used in the current study are applicable to adolescent populations, either in the Introduction or the Methods. Otherwise, it seems like the lower alpha scores on all measures for the Italian sample than the Swedish samples could be due to the fact that the measures were originally developed for adult populations and so are less reliable when applied to younger individuals. Having said that, the alpha scores are not in any cases low enough to cause any real concern for the current findings.


General minor points:
Line 96: It might be a bit strong to say that the subjective wellbeing constructs “lead to” higher psychological wellbeing given that none of the studies referenced in support of this claim appear to have used longitudinal samples/methods

Lines 261-267: It seems a bit beyond the scope of the current study to suggest harmony as a potential social component of wellbeing and then effectively dismiss its use in further studies due to not being social enough. It might give more focus here to simply recommend using another measure that the authors are more satisfied with as a potential social index of wellbeing.

Reviewer 2 ·

Basic reporting

In terms of basic reporting the article meets all criteria. Written in clear and unambiguous English. Clear introduction and background. Relevant literature and appropriate figures submitted. Self-contained and raw data appended.

Experimental design

In terms of the design I would query whether the article ultimately addresses the question posed in the summary and abstract. Having addressed the bearing of religious and cultural context on the conceptualisation of the ideal life, the study did not ultimately address this question. No measures of religious belief or participation were administered, nor was the construction of ideals addressed. Rather the authors conducted a comparative SEM design to look at well-established empirical predictors of psychological wellbeing and how these differ in the two contexts. Significant aspects of the original question and introduction are therefore ignored in the bulk of the methodology.

Validity of the findings

The SEM modelling of the two cultural samples are therefore appropriate for analysing the predictors used in the model.

Additional comments

Overall this is a welcome addition to international cross-cultural work on psychological wellbeing. Given my interest in both positive psychology and psychology of religion I was nevertheless slightly disappointed by the lack of elaboration on cultural/religious and ideal factors covered in the discussion and summary in the study itself. The study tells us that life-satisfaction differs as a predictor in the two cultures but doesn’t offer an answer on how religious variables or ‘ideal life’ constructions relate to this finding. My recommendation would be that either these references in the intro/discussion sections are de-emphasised in accordance with the actual data gathered or alternatively, elaborated speculatively in the discussion section as possible explanations worthy of further exploration not fully addressed here. On the whole though I welcome the research direction and questions addressed by the authors, and view it as a worthy contribution to the field.

---

## Round 0.2 · accepted · Accept

All the comments raised by the reviewers have been addressed.